# Leveraging Smart Contract in Project Procurement through DLT to Gain Sustainable Competitive Advantages

Evin Özkan [1], Neda Azizi [2],* and Omid Haass [1]

1   School of Property, Construction and Project Management, RMIT University, Melbourne, VIC 3000, Australia; evinn.ozkan@gmail.com (E.Ö.); omid.haass@rmit.edu.au (O.H.)
2   School of Business, Torrens University Australia, Melbourne, VIC 3000, Australia
*   Correspondence: neda.azizi@torrens.edu.au

**Abstract:** Project delivery on time, with agreed quality and assigned budget, is the desire of project-based companies. Time, quality, and cost are determinants of project success; however, organisations suffer from achieving these three success criteria at the same time. Failures in project delivery cause the loss of the competitive advantage. The recent digital technologies introduce smart contracts to supply chain (SC) operations for improving SC processes. Project procurement is the area for smart contract implementation to deliver successful projects and gain sustainable competitive advantages. The aim of this study was in explaining how smart contracts benefit project organisations through project procurement. Qualitative research design guided this research with phenomenology. Semi-structured interviews generated the data. The obtained research data were analysed with thematic, textual, and discourse analysis. Published industry reports were used to triangulate the data. This study demonstrated an integrated relationship model to answer the research question. The research findings initially identified the fact that smart contracts improve procurement efficiency through cost, time, and quality. Secondly, smart contracts build a trust-less platform where reliability is delivered and reinforced with transparency, traceability, and security. This study found that enhanced procurement efficiency and reliability meet requirements to gain sustainable competitive advantages. This study intends to contribute to industry practices and future research. The correlation of project procurement management success, smart contracts, and sustainable competitive advantages are expected to guide feature research and business practices.

**Keywords:** project procurement management; smart contract; competitive advantage; qualitative research; sustainability

## 1. Introduction

Project management is an approach which is becoming more and more important for today's businesses. The Project Management Institute (PMI) [1] defines project management as a way of creating value and staying competitive in the business environment. The role of project management comes into prominence in industries such as construction, information technologies (IT), and manufacturing [2]. The project management body of knowledge correlates with 10 knowledge areas, with procurement management being one of them [1]. The challenge for project managers is to deliver successful projects whilst facing numerous risks [2].

The global project success rate is surveyed by the Project Management Institute (PMI) [3]. According to the results, less than 60% of analysed projects were delivered within the projected budget, and the rate of project completion on time remained slightly above 50%. The total project cost and time vary in parallel with the efficiency in the procurement process due to its association with payment and logistics activities. To give an example, from time to time, project managers might face material acquisition challenges such as payment, product misfit or raw material, product/service prices, delivery time,

contract management, and supplier relationship. The more efficient the procurement process a project has, the less money and time lost during a project.

Companies aim to accomplish the procurement operations at the lowest price and within the shortest time because procurement conditions influence the cost and delivery time of the output [4]. In project-based industries, meeting key project performance indicators (KPI) including the project delivery time, quality, and cost helps to satisfy customers. Regardless of the industry and business size, every firm is in a competitive environment. Thus, companies that offer the most convenient solution to their clients are likely to forge ahead and gain competitive advantages [5]. Attaining competitive advantages in the industry support enterprises to generate more profit [6].

Recently, electronic procurement (e-procurement) has taken its place to increase efficiency thanks to the Internet. Development of computer and internet technologies forces companies to adopt and adjust traditional processes to digital versions. In this context, businesses combine software to follow-up operations. The primary goal is enhancing the efficiency of procurement activities, which results in a cost-effective and shorter procurement process. Investing in information technologies is a strategic step to align business data, raise productivity, and expedite clients' demand fulfilment, maximising profitability [7]. Nonetheless, the Internet can be harmful as much as is convenient. Accordingly, software developers have worked on new applications to launch well-protected internet solutions. In line with these efforts, smart contracts and Blockchain were invented. These technologies have provided the opportunity to make secured, rapid, and cheap transactions [8].

Project procurement management, DLT-based smart contract technology, and competitive advantage are major concepts that bring companies significant added values. When the abovementioned information is taken into consideration, gathering these concepts (Blockchain technologies, project procurement and, competitive advantage) may provide a promising solution to project-based companies. These particular types of project-based companies can improve the existing project procurement process by investing in DLT-based smart contract. As a result of this investment, they would be able to attain competitive advantages and contribute to the power of the businesses. In the existing literature, smart contract technology is associated with financial services and supply chain management; however, the previous research did not focus on how smart contracts can facilitate project procurement [9,10].

The importance of procurement activities in delivering a successful project and gaining competitive advantage, as well as the strong match between the mentioned smart contract features and procurement process's shortcomings. In essence, vulnerability, lack of security and information flow, and the gap in the literature about this subject, are necessary to research. Undoubtedly, this study is important and value-added by reason of its contribution to the literature and business environment. Smart contract technology is quite fresh and not well-searched [8]. This research will bring a different dimension into the literature around emerging Blockchain and smart contract concepts. Furthermore, this research will provide an answer to the project-oriented businesses for the questions of how they can gain sustainable project success and competitive advantages. The sustainable competitive advantage is related to Porter's five forces strategy [11], and, obviously, the smart contract technology is a promising investment to meet this model by proposing procurement activities improvement, overall activity time, and cost reduction, as well as operational efficiency and transparency augmentation. The existing literature does not demonstrate the relationship between smart contract, project procurement, and competitive advantage concepts. Thereby, we outline the research question as follows: How do enterprises gain competitive advantages in project procurement processes?

This paper has multiple objectives, as listed below:

1. To criticise current problems businesses experience in the project procurement process. Moreover, to contextualise project procurement by mentioning its importance in overall project success and gained competitive advantages.

2. To examine how Blockchain and smart contract technologies are utilised to overcome problems in project procurement, as well as the benefits they provide to enhance the procurement process in a project.

3. To analyse how these improvements support companies to gain competitive advantages in their industries.

4. To add a new dimension to the extant literature.

This study applied qualitative research methods towards exploring and understanding features of smart contract and Blockchain technologies that can facilitate competitive advantages gaining through project procurement management. The research adopted interpretivism (constructivism) as a research paradigm. The engaged research strategy is phenomenology in association with characteristics of constructivist paradigm, which are embracing multiple realities and constructing new knowledge. Phenomenology leads researchers to reveal an individual's experiences in a specific situation via unstructured interviews, observation, or documents [10,12]. The selected research method was semi-structured interviews with 5 to 25 experts in Blockchain, smart contract, and project procurement fields. Interviews took place in different ways, namely, face-to-face, online, and focus groups. Additionally, we aimed to obtain secondary data through early research and case studies in order to provide a supportive data set to strengthen the research question's answer. Finally, the gained information moved forward through the data organisation, compiling, coding, restructuring, and interpreting steps to arrive at an answer.

The remainder of this paper is structured as follows. The next section presents the research background, followed by an introduction to the concepts of project procurement management, competitive advantage, and smart contract. The research method and details of our approach to data collection and analysis are then presented. Next, we present the research findings, and the paper concludes with an investigation of the theoretical and practical contributions, research limitations, and recommendations for future research.

## 2. Literature Review and Theoretical Background

The review of the literature comprised academic publications about three principal concepts (project procurement management, competitive advantage, and smart contract) that were the foundation of this research. The literature review aimed to demonstrate if firms provide a competitive advantage by using smart contract and distributed ledger technology in procurement operations in a project environment.

### 2.1. Project Procurement Management

Procurement is supplying necessary goods or services for business activities via external suppliers. Independent from the project complexity and project type, each project requires material and service purchasing. Procurement management takes a crucial role in project-based businesses [13]. Under the project management methodology, procurement management is a set of operations that contains planning procurement process, conducting and controlling procurement activities [14]. Ansah et al. [15] emphasised that industries which require a project management approach, such as construction, put procurement in the centre due to the high necessity of materials and services to run a project. Therefore, the performance of the procurement process has a determinant impact on the project outcome [16].

Procurement activities are inevitable in project-based businesses. These activities might be simple or complex and recurring. The more complexity a project procurement involves, the more enhanced tools, techniques, and skills a process demands. Project procurement is one of the areas which is closely linked with project success [17]. Usually, project success is assessed by three factors, namely, cost, time, and scope. Project procurement method and management have a significant impact on the total project cost and duration [18]. This points out this procurement management determines the overall project success. Moreover, well-managed procurement process enables project owners to decrease project expenses by providing the opportunity of reducing paperwork cost and increasing

cash flow efficiency and supplier quality [19]. Reduction of project expenses stimulates project profit growth.

The significance of the procurement process raises the importance of challenges. Project managers and companies face various challenges during procurement activities. As highlighted by Owusu et al. [20], procurement activities are vulnerable to corruption, particularly public procurement in infrastructure projects. Their further statements explain the reasons for this vulnerability of public procurement. The complexity of the process and the vast number of participants make the process hard to manage, while favouritism between counterparties based on political, economic, and personal relations opens the process to abuse. The absence of transparency and accountability may give rise to unethical behaviours. They analysed the corruption in procurement by dividing the procurement process into three phases, pre-contract, contract/contract administration, and post-contract. The result of their study unveils that the most vulnerable stage is contract/contract administration. Post-contract and pre-contract stages follow, respectively.

Information technologies and the Internet enable organisations to digitalise their manual procurement processes. Governments have started integrating e-procurement into their traditional public procurement phases to acquire higher transparency, take advantage of increased competition, and uplift the procurement efficiency [21]. Information and communication technology (ICT) facilitates business process digitalisation and automation [22]. Purchasing requires both internal and external connections. E-procurement enables users to share information with external suppliers via the Internet [23]. This automated information sharing feature of e-procurement helps corporates to sustain accuracy at a higher level. However, security issues on the Internet such as hacking overshadow e-procurement's advantages.

Klaus-Rosińska and Iwko [19] define procurement as a process that is composed of identifying the demand, placing the order, approving the purchase, and paying the invoice steps which are parts of the supply chain management (SCM). Supply chain management means the process that includes the flow of information and materials from suppliers to end-users [24]. However, the larger the scale a process has, the higher the risk of defaults and delays emerge [25]. To maintain the SCM success, authorities need to pay attention to the quality of the information flow [9].

### 2.2. Competitive Advantage

Competition is an integral part of the business environment. Firms, naturally, target to attract customers and earn profit from them. Nevertheless, this common intention also raises the competition among industry participants. To obtain an advantageous position among rivals, companies need to understand the competitive advantage concept and how to apply it in their businesses. Competitive advantages are linked with the business strategy. Therefore, enterprises should, initially, develop a strategy and then execute it. The approach they can hire changes according to the company and industry structure, as well as rivals' actions. Due to that, companies need to analyse their internal sources and external factors. Porter and Marshall [11] built a theory on competitive advantages. Cost leadership, differentiation, and focus are three ways of gaining competitive advantages.

Figure 1 illustrates the generic strategy matrix. Cost leadership refers to the accomplishment of a value chain at a lower cost compared to an equal value chain carried out by rivals. Procurement activities have an influential role in achieving and sustaining cost leadership. Differentiation is the second type of competitive advantage strategy, which means gaining a unique position in the industry by restructuring operational activities or the value chain. Differentiation covers a wide range of alternatives, for example, product/service-oriented changes, technology upgrades or changes, and human resource-related changes. The major disadvantage of differentiation is high implementation costs. The third approach, focus, explains becoming stronger in a narrow target market either as a cost leader or differentiator. Besides these three main strategies, companies can adopt a hybrid strategy, that is, a combination of cost leadership and differentiation [26].

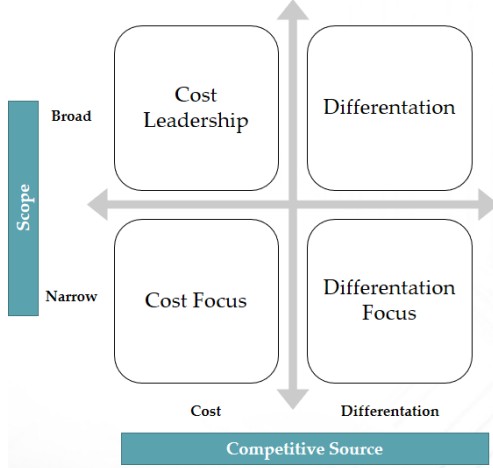

**Figure 1.** Strategy matrix.

Porter [27] is the strategic management guru associated with the competitive advantage of the performance of carried-out business activities. He argued that achieving operational effectiveness in core business activities (creating, producing, selling, and delivering products) is an indicator of performing better than competitors. Enterprises may make investments in increasing performance of business operations. Hence, maintaining the gained competitive advantages must be the second goal for companies after gaining them. The five competitive forces model defined by Michael E. Porter [28] guides companies to succeed in these goals. This model assumes that the factors listed below are five determinants of competition in a market. Having a bargaining power over suppliers is essential to perform better during the procurement operations in a project.

- The threat of new entrants: Easiness of market entrance prompts existing players to lessen the profit or make an investment in differentiation to stay competitive.
- Power of suppliers: Suppliers' power is an important parameter due to the determinant effect of procurement activities on the product price. Thus, industry participants need to be capable of limiting suppliers' power in different ways such as being eligible to switch the supplier and renewing the product to break the dependence on a specific supplier.
- Power of buyers: If customers are more powerful than the company, the company has to obey rules set by customers. To flip this, firms should offer a unique product or value.
- The threat of substitutes: Replication of a product stimulates the price-oriented competition in a market. It is advised that companies provide products or services that are difficult to copy.
- The severity of rivalry: Intensity of competition might expand a market, increase innovation, or deteriorate profitability. It depends on the type of competition among rivals.

### 2.3. Smart Contract

A smart contract has varied definitions in academia. In the most general sense, a smart contract is a digital application which is built on the Blockchain technology and carries terms of a contract in a digitally coded version [9,29–31]. Although the smart contract concept was launched by Nick Szabo in 1997, until Blockchain technology's discovery, the smart contract did not have a supportive technology [32]. The new phenomenon of the Internet, Blockchain, is the underpinned technology of different cryptocurrencies, whereas the well-known types are Bitcoin and Ethereum.

Blockchain and smart contract gained their popularity when Satoshi Nakamoto announced Bitcoin in 2008. The main reason for conceptualising Bitcoin was in developing an alternative payment method which offers solutions for incompetencies of traditional

trust-based payment methods [33,34]. They recommended a new financial instrument that can be used between counterparties as a tool of payment. Besides these factors, Bitcoin is a way of excluding financial and legal intermediaries from and adopting a proof-based operated cryptocurrency in peer-to-peer (P2P) interactions [32].

Ethereum has an enhanced technology compared to Bitcoin and it is much more suitable for use different applications [34]. These two digital tokens underpin the smart contract digital application. "Ethereum is one of the most popular decentralised platforms for smart contract applications" [22]. Tso et al. [35] provide the main reason for this by referring to Ethereum's characteristics. They defined Ethereum as more flexible and as an improved version of Bitcoin; hence, Ethereum allows users to transact on both private and public ledgers. Moreover, Ethereum works with Truing-complete (programming language), which makes it a popular coin in the smart contract usage [36,37].

The most important difference to remember is that blockchain is just one type of distributed ledger. Although blockchain is a sequence of blocks, distributed ledgers do not require such a chain. Furthermore, distributed ledgers do not need proof of work and offer–theoretically–better scaling options. In the literature, scholars have emphasised the improved information storage and data security that Blockchain offers [9,31,38,39]. Blockchain-sourced transactions have a unique chronological code [9] and cryptography that store and secure the data permanently [39]. Nofer et al. [34] explain how the security and verification systems work as follows. The verification system assigns unique random hash values with timestamps to each block of a chain in order to avoid possible data tampering and frauds. This high-level data protection system decreases the likelihood of an internal and external data manipulation.

Francisco and Swanson [38] indicated that every detail of an interaction between two peers (P2P) is replicated and forwarded to decentralised shared ledgers. All participants of the transaction are able to monitor the related data without the capability of making any change on the very same record [34]. Moreover, duplicated records go through the cross-checking step to ensure that the records are still the same as their first version and matching each other [40,41]. This function of the Blockchain increases transparency and reliability whilst it obviates detrimental issues generated by unethical events that might occur in professional relations [37,38].

As the Blockchain implementation and usage are becoming widespread, the nature of monetary transactions would change [34]. Additionally, it has been stated that smart contract and Blockchain concepts can be a platform to carry out P2P transaction not only on the basis of money but also on different tangible assets and properties. The verification system of blocks and transparency of distributed ledgers create a trust-based relationship which allows for the exchange of tangible and intangible assets between peers, even though they have not had an interaction yet [22,41].

On the other hand, DLT supports companies to reduce the number of intermediaries and auditors involved in a contract [38]. Including intermediaries in contracts means sharing even the key information with third parties. This is highly contradicted considering information security. In addition to security concerns, leaving intermediaries out of contracting or transactions is more than likely to decrease the total project cost and duration [34]. In project-based companies, project delivery within the planned time and cost is vital in terms of speaking about project success. Therefore, Blockchain technology is expected to make a contribution to companies' competitive advantages by shortening the estimated project duration and being cost-effective.

Although Blockchain and smart contracts are generally associated with financial services and the finance sector, Francisco and Swanson [34] and Liu et al. [39] claimed that Blockchain technology has a power of changing other business industries. Alongside finance, researchers have predominantly investigated smart contract's benefits in supply chain management (SCM). Discussing Blockchain's features in detail would be helpful so as to have a better understanding of the encouraging improvements in SCM processes

after smart contract implementation because smart contracts are expected to provide all the advantages that Blockchain brings [42].

Chang et al. [22] discuss the low efficiency of the current supply chain processes. First of all, they indicated the poor synchronisation and the coordination problems in particular as the root causes of low efficiency. The presence of a smart contract enables companies to access and monitor the immediate information. The supply chain process is based upon an information flow between a large number of players [41,42]. Accelerating the information flow means reducing the processing time, which contributes to the project completion time. Secondly, as mentioned in the same article, enhancement in the process synchronisation and automation level are outputs of the smart contract usage. Finally, the smart contract solution might incline engineering companies to reassess and redesign the existing business processes (business process redesign—BPR) for a better adaptation. To sum up, all abovementioned developments would shorten the project duration, meanwhile increasing the efficiency and bringing competitive advantages.

Along the project procurement process, the project team needs to go through a bidding process to find the most suitable product at a reasonable price. Tso et al. [35] researched smart contract usage in electronic bidding (e-bidding). Their compelling findings are trust-related issues, high operation costs, and resource requirements of the bidding process. Although e-bidding is becoming more common, this method still has shortcomings, for instance, privacy, safety, and reliability. They propose smart contracts using Ethereum due to the characteristic of them. Relying on the abovementioned Blockchain features, e-bidding on the smart contract would satisfy confidentiality and security criteria. Furthermore, the Blockchain removes third-party attendants, thus cost, time, and resource savings emerge. Entire bidding steps are taken in accordance with rules and regulations that are coded into the smart contract; hence, ineligible bidders are eliminated and cannot proceed to the next steps. As a result, fair approach towards bidders plus time and cost savings support the project owner to reflect corporate reliability that attracts buyers and suppliers and strengthens competitiveness among its opponents [37,43].

Smart contracts consist of digitalised agreements and regulations assigned between the contracting parties [40]. Users can design and customise their own contracts (Chang et al., 2019). The contract is self-executed [40], and, when the required conditions are met, the particular transaction is carried out [9,31]. Punctuality of business transaction promotes reliability and well-structured contractor relationship. Francisco and Swanson [38] note the positive correlation between the corporate image and reliability of suppliers during the project execution.

Liu et al. [39] analysed the smart contract usage in the construction industry, which is one of the project-oriented industries. Poor management of payments, liabilities, cash flow, and liquidity endanger the existence of a construction company. One of the major problems of the construction industry is financial management due to its relationship with payment operations and business sustainability. Smart contract application brings forward contract and payment automation. In this way, projected payments are secured, and late payment issues are precluded [33,44]. Liu et al. [39] expect that this technologic advancement in particular processes would have profound effects on both construction companies and the construction industry. Another large problem of the construction companies is to meet regulation and compliance. A smart contract can assess, audit, and verify the project data with the help of immutable block records.

Ahmadisheykhsarmast and Sonmez [45] developed a smart contract payment security system named SMTSEC to eliminate or reduce payment issues in the construction sector. They developed SMTSEC through an automated computerised protocol which runs on a decentralised blockchain and tested the system on a real construction project. The case that has been chosen to implement the SMTSEC was a powerhouse building in Turkey.

Traceability of goods has become more and more important; hence, companies which pay more attention to providing this information will gain competitive advantages over their rivals. In today's world, the Internet and internet technologies enable people to

reach a broad range of information. Accessibility of information leads consumers to learn more about products they intend to purchase. Wang et al. [31] claim that products, especially those which have a complex supply chain process, demand a well-designed activity tracking system, such as smart contracts [46]. According to their study and the implemented sample system, smart contracts support process traceability, decentralisation, accountability, system scalability, and data security.

Even though the DLT-based smart contract application is convenient for project-based companies to increase project procurement efficiency, it has drawbacks in different fields [6]. Chang et al. [22] imply a set of imperfections for which smart contract and Blockchain technology need still time to develop. "These challenges include legal issues, lack of standards and protocols, privacy issues, and error intolerance" [22]. According to Liu et al. [39], the key challenges engaged with a smart contract are high priced licence fees and implementation cost. Chen et al. [9] mentioned the security problems by stating that the cryptographic security system is not capable enough to prevent the data from internal and external cyber-attacks. He advocates the existence of risks on the Internet.

He et al. [47] investigated various security concerns among industry and academia with regards to application of Ethereum-based smart contract technology. They have compared several mainstream audit tools from various perspectives such as manual audit, the existing audit tools including Oyente, Mythril, and Porosity. The proposed security enhancement solution was based on combining dynamic auditing with static auditing and proposing different dynamic and static auditing methods for various vulnerabilities.

The business world is a competition-intensive environment with a great number of firms' participation. Every step and investment of companies would bring about a detrimental or successful conclusion. Competitors in the same business field have an advantage over others as long as they engage in at least one of these concepts, cost leadership or differentiation [11]. In the project management field, procurement is an inevitable phase and, also, it has a direct link to cost advantageous and differentiation concepts to maintain competitive advantage plus the total project cost and time to ensure the project success. Over the procurement process, companies experience several problems such as unethical operations, issues with suppliers, and ill-managed procurement processes. Developments in computer technologies grew Blockchain and smart contract platforms [46]. Blockchain empowers business processes to become more transparent, traceable, secure, decentralised, and fast. A smart contract is one of the Blockchain applications; therefore, it is equipped with Blockchain' features. In order for the information gained from the literature to be summarised and combined, a DLT-based smart contract implementation can contribute to companies' competitive advantage through fixing shortcomings of project procurement and shortening the operation time and cost.

*2.4. Summary*

The literature review aims to demonstrate if firms provide a competitive advantage by using smart contract and distributed ledger technology in procurement operations in a project environment. To conceptualise this, firstly, procurement management and its importance were researched and expressed. Procurement management is one of the 10 knowledge areas of project management methodology. The success of a procurement process is an indicator of the overall project success. In general meaning, procurement consists of searching for suppliers, contracting, and payments. As demonstrated in PMI's [3] survey, businesses are experiencing difficulties in terms of managing project and meeting project success criteria. Nearly half of the projects exceed the original budget and the projected duration. The ongoing problems cause extra costs, corporate image damages, and customer dissatisfaction. Improvements in procurement management would decrease the frequency of delays and cost overrun.

Secondly, competitive advantage was articulated regarding this study. The literature review uncovered the procurement-related problems which companies are suffering from. These problems inhibit businesses to maintain the competition even to survive. The

literature on the competitive advantage suggests that businesses can build strategies through either the product price, value chain, or both of them. Moreover, Porter's five forces model advises businesses to stay competitive by holding the bargaining power on their hand.

Thirdly, the new phenomenon Blockchain-based smart contract was introduced. Smart contract was found at the end of the 1990s; however, the concept was fresh and the current technology at that time could not support it. In 2008, the first cryptocurrency, Bitcoin, was developed, providing an opportunity to use smart contract in businesses. Blockchain empowers the high-level data security and transparency, which increases the reliability and traceability. Transaction data are sent to share distributed ledgers to be stored. The verification system of Blockchain does not allow for data tampering. Moreover, the smart contract has a self-executed characteristic. After recognising particular agreement conditions and policies, transactions take place when prerequisites are met. In this way, the parties can eliminate intermediaries and trusted third parties from the procurement process. Additionally, smart contract and Blockchain deliver process automation. Payment orders are finalised between contractors autonomously, and this certainly would make a significant contribution to the professional relationship.

As a consequence, a smart contract application through DLT is considered to return momentous advantages in project procurement management. Smart contracts are expected to help project-based organisations to gain significant competitive advantages in the light of project success criteria. It fixes data security and verification problems generating from the Internet, reduces the number of external participants in a contract, and enables automated process execution and process monitoring. In the corporate viewpoint, these benefits reinforce the cost-effective, reliable, high quality, and on-time project delivery.

## 3. Research Design and Methods

In this research project, the aim was to explore and understand how DLT-based smart contract usage in project procurement contributes to gain of competitive advantages in project-oriented businesses. The research aim points out the requirements of answering a question which starts with "how", instead of testing a hypothesis. Exploration and investigation of a concept to understand it well are strengths of qualitative research. Besides these, external research limitations generated by policies and rules, constraints related to the research duration and budget, possible difficulties in counting a large number of samples in the data collection phase address the qualitative research approach. This research was conducted by employing the qualitative research and the following parts for exploring this methodology deeply.

### 3.1. Data Collection and Sampling

The required data type is qualitative, in other words, verbal data. Denzin and Lincoln [48] argued that the data are available in human and non-human sources, and researchers can obtain the data from different sources and with different methods. Phenomenology advises researchers to collect the data via in-depth interviews, documents, and observations [48]. The researchers recommend counting 5 to 25 interviewees in the data collection process involving five case studies. Interviews are crucial to attaining primary data. Interviews were conducted face-to-face and online specifically for this project. If the necessary permissions and arrangements, such as the common time and venue, were to be finalised, focus group interviews would be a useful data collection method. Focus group interviews have an opportunity to promote the richness of the collected information in association with the group discussion [49]. The question type is open-ended, which works coherently with an in-depth interview and qualitative research. All interviews were conducted according to the rules and regulations mentioned in human research ethics. Sessions took 30 min to an hour and were recorded after receiving permission from interviewees. In addition to primary data generated through semi-structured interviews,

secondary data would support the research. Early research and case studies related to the procurement process and Blockchain can enlighten the research.

The researched sample has a critical impact on the acquired data. Strategically and theoretically built samples support the findings in terms of reliability, generalisability, validity, credibility, and quality [50]. They also contribute to overall research success. The sample consisted of experts who have academic and professional Blockchain and smart contract experience plus knowledge. Accessing and understanding experts' opinions was the main target of this sampling. Considering the intention, we formed the sample purposefully. Experts were searched from different industries and institutes to sustain information diversity and increase the generalisability of the outputs. This study aimed to produce international output. Thus, participants overseas were considered in the target sampling.

*3.2. Data Analysis*

The data analysis method consisted of textual, discourse, and thematic analyses. The interview data came in different languages, namely, Turkish and English. Data analysis programs are not capable enough to analyse the verbal data in different languages. Although there is a variety of qualitative data analysis software, we managed this phase manually. The challenging duty in data analysis was staying natural and not having biases. Hence, we tried to read transcripts without having an opinion but only read the text.

Initially, we applied textual analysis that necessitated us to select a context and narrow the scope down. The context we preferred was the business context regarding our previous studies and experiences, plus the business aspect of the research area. The textual analysis supported us to understand the participants' opinions and experiences in the business context. When we were analysing the data, we read the text in detail and discussed participants' responses within the business context. We used sticky notes to write the interpretation of important words and concepts we found in the text. This way of working gave us the opportunity to combine and compile sticky notes for the comparison of the outcomes of all applied data analysis methods.

Secondly, discourse analysis took place. This method was decisive in this study because of collecting the data in Turkish and English. Therefore, Turkish interviewees' words would have extensive meanings. Even though the other three interviews were given in English, some interviewees came from high-context cultures. As discourse analysis requires, we read the transcripts line by line to extract the deeper meaning of keywords and sentences. Furthermore, to enhance the analysis, we listened and watched the interview files with the purpose of having a better understanding of the tone of voice and body language. As in textual analysis, we used sticky notes for the notetaking.

Thirdly, we analysed the printed interview transcripts and highlighted sentences that addressed the research question. We detected repetitive words and concepts to code the data for the thematic analysis. The review and rearrangement of these codes created two main themes and six sub-themes in relation to the research question. The main themes were efficiency and reliability. The theme efficiency consisted of three sub-themes: time, cost, and quality. Transparency, traceability, and security were sub-themes which formed the theme reliability. The findings generated through the data analysis are presented and discussed in the following sections in the paper. The outline of data presentation and discussion was constructed on the basis of the thematic analysis findings. Outcomes of discourse and textual analysis are reported under the headings for themes and sub-themes.

## 4. Findings and Discussion

The data obtained from documents and five industry experts brought about meaningful results after the data analysis. The findings emerged from this effort in harmony with the findings in the reviewed literature. Experts spotlighted the particular benefits of smart contract and Blockchain in different words and examples. These are as follows:

- Supply chain is one of the most suitable business processes to implement Blockchain and smart contract.
- Supply chain and procurement tackle efficiency problems. Blockchain and smart contract improve the procurement process efficiency.
- Blockchain and smart contract can decrease the procurement process time by eliminating manual jobs.
- Blockchain and smart contract support businesses for the cost reduction and gain competitive advantage through cost advantage strategy.
- The process quality can improve with Blockchain and smart contract usage.
- Reliability problems between project stakeholders can be removed.
- The process transparency can elevate with the advance transaction data and shared database.
- Blockchain can facilitate the process and product traceability in every phase.
- Blockchain can provide improved data security.
- Blockchain and smart contract adaptation can support enterprises to gain competitive advantages.

Expert views pointed out that smart contract can improve project procurement management and facilitate competitive advantages.

Figure 2 demonstrates the categorisation of findings through the thematic analysis. The following subtitles explain how those companies can gain competitive advantages by using a smart contract in project procurement processes through the themes and participants' answers.

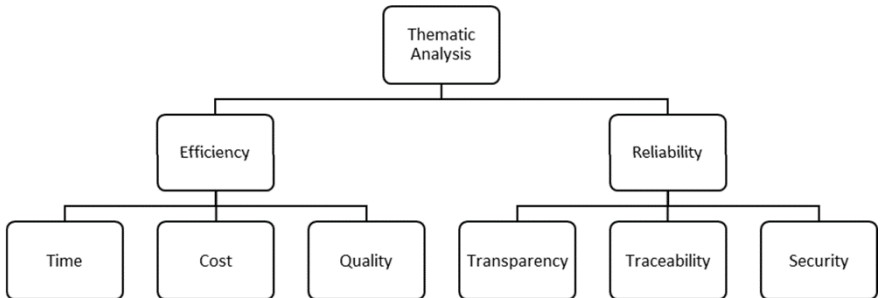

**Figure 2.** Theme structure.

*4.1. Efficiency*

Interviewees stated that Blockchain has a wide usage potential in all business departments. Finance is the pioneer of Blockchain implementation projects in the business environment, whilst the supply chain is preferred. Not only the interview data but also documents and the entire journey indicated that supply chain and procurement need to develop with Blockchain and smart contract implementation.

"Thanks to Bitcoin and other cryptocurrencies, numerous projects have been carried out in finance. Many projects have been embarked especially in the supply chain for the last three years. . . . Recently, governments have initiated projects related to the integration of Blockchain into customs processes in international trade, international transportation and supply chain" (Case 3, Blockchain and smart contract consultant).

"Blockchain is a specific DLT system. There have been projects run on Blockchain in the financial sector. For example, all banks in Australia keep their centralised data in their silos. It will take some time to switch to Blockchain from this level. They tested the system. However, due to bitcoin getting involved in fraud events, they even execute interbank projects through APIs (Application Programming Interface). CBA had an application for money transfer called Beem. The plan was actualising this on Blockchain. Authorities changed their mind because public Blockchain and bitcoin are notorious and have bad perceived security among society even they are secure. In general, the best usage area of Blockchain is in the supply chain" (Case 2, Blockchain and smart contract developer).

Case 3 implied that the process digitalisation is not for only companies but also countries. Case 2, who is a Blockchain and smart contract developer in the banking industry, detailed the usage case of Blockchain in finance with an Australian example. Although Blockchain has a financial aspect regarding the cryptocurrencies, the implementation and practice of this technology requires social support and trust in finance. Moreover, his words showed that companies in finance pay attention to customers' thoughts. They assume a possible resistance from customers to the new system which might deteriorate into competitive advantages.

The supply chain was defined as a complex process that includes numerous parties and operations from different business areas. Interviewees categorised this complicated process as procurement, contracting, invoicing, transportation, and delivery of products. Reports discussed the growing complexity with the participation of global partners. The complexity in a supply chain was linked with posing threats to the success of the process. Case 1, who is an expert in the built environment, observed significant and repetitive problems in the construction supply chain and suggested the Blockchain technology confidently to fix these problems.

"As I said, there are lots of areas we could use Blockchain, but mainly it's a supply chain. There are different ways where you locate supply chain and payment issues, ownership, information integration, separation, and governance. There are lots of issues which can be solved potentially by Blockchain. The construction industry has a lot of problems. People don't know where to make a stand. The late payment is a problem, especially for small-medium businesses. It's a real industry problem. For example, payment or contract management" (Case 1, senior lecturer in construction).

Efficiency was one of the most frequently used words in the interview transcripts. The word efficiency was used for specifying the business process of efficiency, which is the procurement process as a component of supply chain management in this study. This study enabled the acquisition of experts' views on what efficiency is and how a smart contract impacts on it. The research participants described the improved efficiency in various aspects of the procurement process with different examples. Their description of the efficiency contained improved process quality, lowered cost, and reduced time. All interviewees were in agreement that the smart contract adaptation increases procurement efficiency in terms of the quality, cost, and time. Additionally, the secondary data verified the findings of interview data by discussing the improvement in cost, time, and quality.

Case 1 stated the improved process efficiency based on her experience. However, she was implying that the Blockchain concept is too young to install business processes and therefore that it is hard to observe its real impact in the industry.

"The real industry project that we used Blockchain, we have just done a demonstration product. It's a demonstration system. There are some papers which they said they have done some demonstration in the industry. They are recent project paper published by Automation Construction by Chinese academics. . . . And my personal experience says that it increases the transparency definitely. It increased the efficiency for the business process" (Case 1, senior lecturer in construction).

The question was if the smart contract increases efficiency and how. Case 4 linked the process efficiency with the spent cost and time through the general perception among businesses. In the business environment, paperwork and contact numbers symbolise time-consuming jobs. His response advocated that smart contract and Blockchain encourage businesses to decrease the cost and time by proposing solutions for these problems. All participants asserted that smart contract executes processes automatically, and this is how it enables the cost and time reduction. Case 3 listed four areas of improvement which create efficiency in procurement "(1) trust between parties, (2) process acceleration, (3) cutting expenses, and (4) autonomy".

"The promises that it can again. Again, it comes back to the practical challenge. If you think about existing networks and where the inefficiencies are, most of the motivation is around efficiency that saves money. . . . If I come into and say this will make the process

more efficient that means less paperwork. That means fewer points of contact . . . It's also for the betterment of a business process that makes supply chain more efficient and more effective" (Case 4, Blockchain and DLT strategist).

The supplier selection process was mentioned as one of the determinants of the procurement efficiency regarding its impact on the project quality and procurement activity time. The interview analysis uncovered the difficulty of finding the right suppliers and reflected the enhancing role of smart contracts in the supplier selection. Participants agreed on some features of smart contract which help to determine suppliers before the contracting. They defined them as (1) evaluating suppliers on the basis of the coded contract requirements by using the historic and searched data about companies, and (2) sorting the qualified suppliers. Case 4 gave a Walmart example to explain how Walmart built a system on Blockchain to work with suppliers which fulfil Walmart's criteria. Case 2 addressed the criteria for suppliers and automatic filtering of applications on smart contract to select the most suitable suppliers. Case 5 and Case 3 cited the example of IBM–Maersk partnership to create a Blockchain application with the participation of various stakeholders. Furthermore, these examples and definitions indicated a customised procurement process.

" . . . we do not know if this is a good supplier, has a good history, can deliver on time. If we use a Blockchain, Blockchain can create this efficiency. Because we can go and audit all the information about suppliers, logs, and transactions, all the businesses we have dealt with. This will help us to determine the suppliers. This is increasing the efficiency because I do not need to search or email anything. Moreover, there is an automation and standardisation that you can do. When you are using a Blockchain or smart contract, you create that among all the party who are participating in our ecosystem. They have to act according to conditions you have created" (Case 5, Blockchain and smart contract consultant).

"There is a shared DLT or Blockchain database. Companies contributing to this database allow the registered members to use this database for a certain fee (private blocks) or for free (public blocks). During the procurement, organisations will be able to develop the planning and supplier selection phases by choosing the products or services that suit their needs with the help of the provided data" (Case 3, Blockchain and smart contract consultant).

### 4.1.1. Time

This study pointed out that experts think the Blockchain and smart contract technology enable time-saving to companies. Participants discussed the shortened procurement time on the nature of the smart contract technology. The common components stated when defining a smart contract were being a digital contract coded on Blockchain, executing operations based upon completion of prerequisites, and triggering following phases automatically. Participants agreed that automation in the procurement process through smart contract accelerate operations. They added that the more manual a current procurement process is, the more time-saving enterprises will achieve with smart contract. The reason for anticipated time-saving was explained as the fast communication between machines rather than human interaction.

Participants' consensus was that a supply chain process involves multiple parties and manual jobs. Supplier selection, payment, and monitoring the contract conditions exemplified these jobs. They suggested that the digitalisation of these processes through smart contracts can eliminate inefficiencies in procurement. Industry reports recommended smart contracts to bring automation into the supply chain process and increase the operation speed. Self-executed supplier selection and bidding were introduced as a significant time saving-opportunity. Additionally, Case 5 drew attention to the emailing in the business environment. He criticised the wasted time and effort while transacting and pointed this as a shortcoming that can be improved. Moreover, the interview analysis unveiled that some participants believe that increased efficiency and shortened time demand less personnel.

"If the targeted outcome is creating efficiency, then we can prove that how using a smart contract reduces the time of business operations to complete. Procurement is the example. It may reduce the procurement process duration from 10 days to 2 days" (Case 5, Blockchain and smart contract consultant).

"All processes that take a long time and operate manually can be converted to digital. For example, if five people work on a particular job in the workplace, only a person can suffice for the same task. As the execution time of the work decreases, man/day savings can be achieved" (Case 3, Blockchain and smart contract consultant).

Although smart contracts were mentioned by experts as a promising Blockchain application, participants found shortcomings of it. Experts raised particular concerns which need to be taken into account while coding the smart contract. Case 1 noted the possibility of disagreement between parties after the smart contracts take place. Her experiences in the construction industry formed the basis of her consideration. Case 2's point was if the system faces a dispute when the system is running. He implied that in the case of a dispute, if the system does not know how to solve that dispute, the rest of the process would stop. One of the consequences of these cases was expected to be spending extra time on rebuilding the system.

"If the processes are manual, yes, smart contract reduces the required time, but in general, it accelerates the operations since it automates the process. However, not every automated system is accelerated. For example, how would the Blockchain decide when the system faces a dispute? Most likely, a smart contract quickens the supply chain process" (Case 2, Blockchain and smart contract developer).

### 4.1.2. Cost

The findings of the interviews and secondary data pointed out a set of cost advantages that companies can attain. Cost benefits of Blockchain and smart contract were perceived as an attractive outcome of the process digitalisation among participants. Participants articulated these cost advantages from a different perspective. They provided examples to show these benefits on the basis of their experiences and technical features of smart contracts. All interviewees linked the cost savings with improving the procurement efficiency and gaining competitive advantages over rival companies. Moreover, some experts reported negative consequences of the accomplished cost saving through a smart contract.

Four of the five interviewees associated the cost, time, and efficiency. The main argument they brought was that the chain of events started with the process digitalisation. They mentioned that the process of digitalisation shortened the time and elevated the efficiency, then cost-saving followed them as a result of betterments in time and efficiency. In the industry reports, this relationship was created on process improvement and gained cost efficiency. The reason for this was stated by Case 5 as "the dollar value of time". Four participants agreed that smart contract usage decreases human interactions and the manpower demand. These interviewees calculated the cost-saving on the lessened time spent on a task and man force demand. However, Case 4 discussed the adverse effects, namely, laying off and internal resistance.

"Potentially it means likely in the short term, people lose jobs. The smart contract makes the business better, yes, but then you have enormous internal resistance to that in the short-term. Because it says what it means for us, it's problematic" (Case 4, Blockchain and DLT strategist).

The other suggested cost saving was related to removing third parties from the procurement process. Three of five participants acknowledged that smart contract is able to allow companies to eliminate the third party. The third party was identified as all participants who have a mediating role. Lawyers, law offices, intermediaries, and other similar organisations were listed in the third party. Case 3 and Case 5 mentioned "Oracle", which replaces the third party. Case 3 added that Oracle brings the game theory in to maintain a negotiation.

Some participants noted cost advantages that companies can achieve through tendering, bidding, and price offer. These participants indicated that Blockchain and smart contract create a competitive environment for suppliers, and buyers can obtain cheaper prices. Case 4 provided an example of Walmart. Walmart was one of the companies that introduced Blockchain to suppliers and managed supplier selection on the Blockchain application. He implied that this strategy enabled Walmart to control current and candidate suppliers. Additionally, in this way, Walmart broke the power of suppliers.

" . . . So bidding and action can also work for in our favour. Most of the time when you go for pricing, you do not know which supplier charges more. However, with the, Blockchain we can have the transparency of what they charge. They all can see who is charging more. In that way, it can create competitiveness" (Case 5, Blockchain and smart contract consultant).

Participants addressed operational expenses, inventory, and transportation costs. Case 3 and Case 5 explained the cost-saving on the example of Maersk and IBM collaboration. According to them, IBM application allows buyers to optimise the delivery with the data provided by Maersk for transportation. The other cost-saving strategy was built on the inventory level. Moreover, the interview analysis pointed out that transaction fees are cheaper and that conversion rates on Blockchain are in favour of businesses.

"Products that require special storage conditions as the process takes shorter will also require special storage conditions for a shorter period. In this case, the cost will decrease. Another cost-saving is the reduction of the stock amount to be kept. Smart contract and Blockchain can contribute to that" (Case 3, Blockchain and smart contract consultant).

Finally, participants mentioned the challenges of Blockchain and smart contract in the financial aspect. Although the accepted thought was cost-saving, participants detected a few shortcomings. It was expected by participants that the fundamental utilisation of smart contract and Blockchain will occur when cryptocurrencies will be authorised to use in business transactions. Participants agreed that Blockchain and smart contract are not ready yet to complete monetary exchanges due to the lack of regulations. Besides this, Case 1 brought the difficulties of how to evaluate the cost-saving. She stated some expenses which companies need to bear, such as the IT infrastructure improvement, Blockchain investment, and public Blockchain transaction cost. The rest of the interviewees stated that these expenses are not costly or will be covered in the long term.

### 4.1.3. Quality

The improved process quality was interpreted from the interview analysis. Participants implied the quality in various aspects and its contribution to the process efficiency. Interview analysis revealed that quality is enhanced as a result of improved transparency, performance assessment, supplier selection, and technology infrastructure. Some participants stated that "the aim is improving quality but the quality is not the result". The interview data pointed that companies can set their quality standards and maintain them by compelling suppliers to meet the set quality standards. It was implied that from beginning to end, buyers can dominate procurement activities and hold the bargaining power over suppliers.

Data analysis unveiled participants' thoughts that transparency, traceability, performance assessment, and accountability are factors of enhanced quality. Interviewees and the secondary data affirmed the performance assessment and accountability as an output of the increased transparency and traceability, after the smart contract implementation. Moreover, some experts mentioned the performance assessment and review functions of a smart contract which guides buyers to work with better suppliers in the next projects. The quality of the process and project was linked with the selection of the right suppliers.

Finally, the essential component of the quality was defined as data. Interviewees believed that transaction logs and history help businesses to check every step of the procurement process. Therefore, the use of the smart contract with IoT (Internet of Things) device installation was advised. IoT devices were described as sensitive sensors which

collect the related data from the process and transfer to blocks in a Blockchain. Participants implied that the advance transparency and traceability of the entire procurement allow supply chain partners to detect inefficiencies and mistakes immediately.

*4.2. Reliability*

Thematic data analysis generated reliability as one of the mega themes. This term reflected the trust between business partners based on primary and secondary data. The collected data unveiled that trust and reliability are important in business interactions, whereas supply chain partners have a lack of trust. Besides the nature of business relations, deficiencies in transparency, traceability, and security were given reasons for the lack of trust. Data analysis declared that Blockchain can be proposed as a technology to provide a trust-less digital environment for businesses, where they can operate safely and securely.

Payment issues were listed in the problems which deteriorate mutual trust. Although buyers and sellers promise to commit to the agreement, especially in the construction industry, debtor parties would not want to pay. Smart contracts are expected to address payment-related problems. Case 3 noted that smart contracts execute the payment after both parties meet reciprocal commitments. In addition, data analysis uncovered that Blockchain provides validated and verified transaction data that can be trusted and tracked by every party. This feature of the Blockchain was emphasised as key to building the trust between supply chain participants. Participants speculated that the mutual trust between companies brings competitive advantages.

Furthermore, the interview data demonstrated that smart contracts and Blockchain involve trusted parties. All parties would be able to audit all transaction logs at any time. The increased transparency would mitigate risks while creating trust between organisations. The other outcome found in the data was the improved accountability.

"If someone has not done their job, the project state is on hold. Because it still waits for other person to complete. Everyone knows who is delaying it" (Case 5, Blockchain and smart contract consultant).

The traceable data and transparent information sharing reached by Blockchain and smart contract promote accountability. In that way, people are expected to know and carry out their responsibilities and correspondingly trust each other.

4.2.1. Transparency

The complexity of the supply chain was a fact that was declared by all participants and industry reports. Reports stated that the increasing complexity of a supply chain in correlation with globalisation, whilst Case 2 noted that a few parties in a supply chain are much easier in terms of managing the flow of information. The collected data revealed that the main sources of the complexity are multiple party participation, lack of transaction data, and multiple data records. The data analysis pointed out that promoting transparency helps businesses to tackle the complexity problem. Interviewees assured the contribution of Blockchain and smart contract in order to build a transparent supply chain process by way of an advanced transaction data record plus, single, and open database. Industry reports also certified the findings from the interview data. Moreover, industry reports published by KPMG and OECD [51] and Deliotte [52] claimed that transparency stimulates enterprises to comply with the code of conduct and sustainability while operating. These two concepts were mentioned as keys of competitive advantages because they have an impact on purchasing decision.

Experts mentioned the importance and high necessity of data in the business environment. Data transparency was remarked as vital as data. They stated that every step in a supply chain has to be recorded and shared with other participants. The primary and secondary data demonstrated that Blockchain provides a verified, trusted, and single data source to all participants of a process. Experts and documents added the facilitation of auditing beyond creating transparency. Some participants said that the accessibility of the transaction history allows users to review records and audit the process any time they need.

Besides this, they noted the possible enhancements in contractors' performance assessment and accountability. The other benefits of transparency were specified as the fraud detection and bargaining power over suppliers obtained by transparency. The transparent ecosystem was considered to create a competitive environment for candidate suppliers and resulted in significant discounts for buyers.

Although experts and released documents indicated a transparent supply chain process from the beginning to end, they expressed their concerns with the increased transparency. Case 4 implied that the transparency in business is a double-edged sword in regards to competitive advantages. The word expose meant the indefensible and vulnerable business. The European Union Blockchain Observatory and Forum mentioned the transparency-related issues in their report. Liu et al. [39] pointed out that "The data transparency afforded by Blockchain platforms, while useful for managing and securing supply chains, can risk exposing confidential information to competitors."

"When it comes to some of the other things that are transparency-related, people don't want to hear their dirty laundry. . . . Wouldn't it be terrific to be able to track this from a procurement stage ultimately to the delivery stage? The answer is yes, but somewhere along that stage, you might expose the business to the world in a way that makes you less competitive. We might expose something which ultimately you just don't want the world to say" (Case 4, Blockchain and DLT strategist).

### 4.2.2. Traceability

Industry reports indicated some significant business issues caused by the lack of data. The non-traceable supply chain process was declared to be one of these issues. Documents revealed that every industry suffers from not being able to trace their supply chain process in particular. Mining, food, technology, and agriculture were some of the industries that implemented Blockchain technology in the supply chain. These industries reported the improved traceability and quality for raw materials they procured. Furthermore, the data analysis acknowledged the risk mitigation for fraud events. Fraud events were defined as payment-related issues, product authenticity, and crossing fair-trade boundaries in tendering. The improved traceability trough Blockchain was approved by the data analysis. Experts noted that Blockchain and smart contract implementation solves the data deficiency and facilitate traceability.

Traceability was mentioned as an influential component of the competitive advantage in the collected data. It was implied that consumers want to know the history of the products they purchase. Interviewees provided industry examples to discuss the marketing and competitive advantage side of improved traceability. Walmart and Dairy Australia were mentioned as two companies which provide the product history to their customers through Blockchain. Reports also discussed the traceability in the aspect of its contribution to gaining competitive advantage. It was further added that Blockchain and smart contracts are not capable enough to collect data for maintaining traceability. It was advised that IoT devices are essential components to attain the immediate information required for traceability.

### 4.2.3. Security

Security was elaborated from several aspects by interviewees. One of them was the security hired by the nature of Blockchain, and the other was the impact of the security on business relations. Experts and documents highlighted the immutability of Blockchain data in addition to being highly resistant to hacking activities. Case 2 described the security steps in a Blockchain. His explanation also unveiled the demand for the commitment and effort of all users of a Blockchain system. Case 5 and Case 4 confirmed that data security problems are caused by coding errors and user's behaviours.

"There is a public and a private (sent over intranets, not visible on the internet) Blockchain. Data is encrypted by using data encryption technology. However, we are talking about distributed ledgers. Since data is distributed, it is more difficult to secure

it. No matter how many places you have distributed the data, you have to secure all of them. Besides the data, you have to secure the servers and, you need to encrypt the keys you use in encryption. However, this does not mean that the Blockchain is less secure. The disturbance here is the increased security risk and workload. All participants must follow the data protection procedures" (Case 2, Blockchain and smart contract developer).

The data analysis pointed out that Blockchain and smart contract can bring security to financial operations in the procurement phase. In industry reports, the increased financial transaction security was explained by the concept named "know-your-customer" (KYC).

" . . . smart contract can conduct validation and verification checks on the payment method to eliminate fraud" (Case 2, Blockchain and smart contract developer).

In addition to fraud, it was noted that Blockchain provides a secured platform for data sharing between parties. Moreover, the enhanced product security as a result of a transparent and traceable supply chain process was added in the reports".

*4.3. Discussion*

The discussion part aims to interpret the findings of primary and secondary data. In this part, we articulated the research findings by linking them with the reviewed literature. Throughout the presentation of the findings, we noticed the interrelationships between each concept. The big picture demonstrated the correlation of project procurement, competitive advantages, and smart contracts. We structured this part with five subtitles. These subtitles aim to articulate and present relationships between individual concepts.

Project management methodology gathers a variety of disciplines depending on the project area. In the business context, projects incorporate several operation areas such as human resources, finance, and procurement. The existing literature stresses the critical role of the procurement and supply chain in business operations in addition to its inefficiencies. Findings of this study assert inefficiencies in procurement in line with the reviewed literature. As presented in the literature review section, the current operations encountered problems in regards to the ill-managed process. The entire procurement process, from the planning to performance assessment, underperforms. This fact costs businesses their competitiveness. Therein lies efficiency of the procurement and reliability in the business environment.

4.3.1. Smart Contract and Procurement Process Automation

Smart contracts are an application of Blockchain. Smart contracts are coded and digitalised contracts on Blockchain that trigger the following phases by themselves when the required criteria meet [53]. Smart contracts are known for making self-executed processes possible. In the development phase, it requires parties' consensus on the contract terms and conditions. Once the contract is coded, it checks, verifies, and validates the necessary conditions and executes the next operation. Government and non-government organisations started transforming repetitive and manual processes, such as the procurement and supply chain, to digitalised operations with smart contracts. The automation in procurement is expected to improve the supplier selection, contract management payment, and monitoring and performance assessment of suppliers. However, different from the existing literature, this study revealed that the automation of processes poses a threat to people who are in charge of managing these processes. Although the automation of process is minimising human-related risks, it increases the risk of internal resistance.

A dynamic service-level agreement (SLA) enabled by smart contract can be used to manage changes and cancelations in the procurement process. This solution is aligned with the distributed service-level agreement management system of Uriarte et al. [54], which proposed a two level blockchain architecture. According to their proposed framework, at the first level, the smart SLA is transformed into a smart contract that dynamically guides service provisioning, and at the second level, a permission-based blockchain system is developed through collective monitoring entities to create objective measurements for the smart contract assessment.

### 4.3.2. Procurement Efficiency and Smart Contract

Inefficiencies in project procurement generate high operation costs. Project procurement consists of contracts; thus, procurement processes may entail intermediaries who regulate the environment. Smart contracts enable organisations to remove intermediaries from the procurement process alongside process automation [38]. The findings of the research matches with the literature. Third-party removal (intermediaries and auditors) ensures cost-saving. Furthermore, the reviewed studies mentioned several possible outcomes of the smart contract implementation. However, they did not state the benefit that project-based companies can derive from bidding and tendering by catalysing the competitiveness among suppliers [43]. The transparency adapting with smart contracts can work in favour of buyers. The transparent ecosystem empowers buyers to control the purchasing conditions and weaken suppliers and contractors. The exchange of the power supports buyers with better price offers. Moreover, ecosystems created by the co-operation of large enterprises (for example, the Blockchain application constructed by IBM and Maersk) allow buyers to optimise the procurement and logistic processes. Another cost-saving opportunity is observed as the reduced human resources. Even though it has a huge debate on its impact, switching manpower to smart contracts brings a significant cost saving. Finally, smart contracts contribute to the planning and inventory management. A well-planned procurement process can save companies to bear the cost of the material shortage and inventory costs.

One of the significant reasons for the operation cost in procurement is high resource requirements [35]. Time is a vital resource for companies. Hence, processing time is one of the major concerns in the business world. Enterprises have a target to shorten the operation time. This goal is an indicator of the improved process efficiency and effective usage of resources. The automation in the procurement process through smart contracts would bring advantages to decrease the processing time [22]. Smart contracts enable communication between machines. The interpretation of findings indicates that communication between machines is much faster than human interactions. Furthermore, supplier selection, monitoring the contractors' performance, and project progress have become repetitive and time-consuming. Automation and digitalisation of these and other similar processes with smart contracts can provide better usage of resources. The decrease in the time allocated these jobs to also reduce operating costs.

Smart contract usage can promote the procurement process quality, while smart contracts influence overall project quality. Procurement process quality hinges upon the information transparency, traceability, and data quality. IoT (Internet of Things) devices are useful tools to collect more accurate data and expand the process database. IoT devices stimulate smart contracts to perform better and enhance embedded autonomy. Paliwal et al. [55] pointed out the correlation between improved data quality and Blockchain adaptation. Their other significant finding is that Blockchain facilitates procured materials' quality [55]. This study generated results in compliance with the literature and added that supplier selection is also a factor to improve quality. Smart contracts automate the supplier selection process by filtering and evaluating candidates. Additionally, smart contracts entitle project owners to monitor the real-time contractor performance and increase the accountability level of contractors. In this way, project owners can accomplish a superior project quality. On the other hand, the findings imply the cost-saving as a result of improved procurement and project quality.

### 4.3.3. Procurement Efficiency and Competitive Advantage

Efficiency means betterment in the business process while it describes the cost-saving and reduced errors in the procurement process [56]. Although in the literature, the procurement efficiency has two parameters, this study suggests that time, cost, and quality, which is called the iron triangle, as parameters to assess the procurement efficiency. These three factors are critical to evaluating the overall project success; therefore, there is a link between the project success and project procurement for project-based companies. Additionally,

the strategy matrix (Figure 1) created by Porter and Marshall [11] is a concept that we can relate to the success in the strategy execution to procurement performance. Businesses need to reduce the cost through the procurement to apply the cost leadership strategy. The differentiation strategy requires an added value to the value chain. Companies can achieve it again through the procurement. Process and product quality may assist organisations in differentiating their value chain, depending on the product and services. These relationships prove that increased procurement efficiency leads companies to gain competitive advantages by way of contributing to the strategy implementation.

### 4.3.4. Reliability and Smart Contract

Businesses that are collaborating seek mutual trust. However, companies struggle to build trusted relationships. Project procurement demands internal and external participation. Since project management has a fragile and complex nature, procurement operations take a critical role in delivering successful projects. The extant literature mentioned devastating deficiencies in procurement. Corruption and misconduct [20], data security, and the high risk of defaults related to the process complexity [25] are major problems. The lack of transparency and traceability in the current procurement process are identified as sources of these problems.

Project procurement has similarities with the supply chain. Supply chain management relies on information flow between parties. Supply chain summarises flows of information and products between buyers and suppliers [24]. Therefore, data quality, transparency, and traceability have a crucial impact on the supply chain success and project procurement. Blockchain is a promising concept with respect to transaction transparency, decentralisation, reliability [39], traceability, security, quickness [38], accuracy, and efficiency [40]. The research findings provide parallel results with previous studies in the context of procurement problems and the power of smart contracts and Blockchain technologies to resolve them. Smart contracts produce verified and validated data. Smart contracts collect the verified and validated data in a single database to maintain a single source of truth. This technology saves collaborating organisations from multiple databases and lack of trust. The other gained benefit is realising accurate and quick information sharing to improve the procurement process.

As referred to in the literature, the procurement process can be easily manipulated. This study found major manipulation in procurement process that, which can cause ethical and legal concerns. Payment is an object of concern in the procurement procedure. Payment-related problems include fraud and missed/delayed payments. When a contract is signed, parties promise for that commitment. Therefore, it is expected that both sides meet terms and conditions as identified in a contract. Besides this, favouritism in terms of supplier selection is an example of misconduct [20] and goes against the fair-trade agreements. The study stresses that the embedded transparency and traceability in the procurement process by smart contracts simplify detection, proof, and prevention of unethical events. The automated supplier selection phase executed by smart contracts can limit external manipulations and treat each candidate equally.

Business data are vital because it includes confidential information. Leakage of this information would cause a loss of competitive advantages. Security concerns are noticeable as aforementioned matters. The angles of security are interpreted as data security and security in the business ecosystem. Data security emphasises the lack of trust in online business tools. The internet poses numerous risks to its users. Public and private organisations digitalise the procurement operation on e-procurement; however, it does not offer a secure environment to them. Thus, Bologa et al. [57] developed a security model designed with an encryption system. Thanks to Blockchain technology, data are secured with encryption, and it provides more reliable data protection. The other feature of Blockchain that supports data security is enabling users to transact on both public and private blocks. In addition, the third-party involvement (auditors, intermediaries, and so on) may necessitate the sharing of confidential information. Smart contract eliminates the third parties and

avoids information sharing. Since the encryption system and private Blockchain upgrade information security, businesses also target to save themselves external security threats. Fraud events (fake product, payment, and misconduct) might become challenging for public and private organisations. The transparent and traceable procurement process can overcome these challenges.

### 4.3.5. Reliability and Competitive Advantage

This study points out the strong bonds between reliability and competitive advantages. The findings define this relationship with Porter's five forces model. Initially, the five forces model by Porter [28] recommends businesses to hold the bargaining power over suppliers and customers. The industry examples given in the interviews and industry reports declare that Blockchain empowers buyers to gain competitive advantages among their rivals through Porter's five forces model. Smart contracts and Blockchain projects in use have proven the increase in customer trust alongside supplier trust. The interpretation of findings revealed other insightful connections. After several scandals occurred in the world (for example, The Rana Plaza disaster, Apple factory in China, and Nestlé's misconduct), consumers began to wonder how their products are produced. Customers who pay attention to sustainable sourcing tend to trace the product journey and prefer to buy ethically and responsibly sourced products [58]. Blockchain and smart contracts support companies to provide product journey and gain competitive advantages. Secondly, the literature and findings indicate the trust-less business platform. Fraud and misconduct damage the supplier and buyer relationship. The transparent, traceable, and verified data generated by smart contracts protect businesses from going against fair-trade agreements or facing unethical events. The nature of smart contracts and Blockchain ensures participants' reliability and security of the business environment [38]. The long-term and healthy relationship between parties create trust and provide competitive advantages.

### 4.4. Summary

The researchers acquired primary data through interviews and secondary data from published industry reports to verify the primary data. The presented findings exhibited links between smart contract, competitive advantages, procurement process efficiency, and reliability. Strong interlinks between individual concepts enabled us to connect them and address the research question. Figure 3 illustrates the discovered relationships. The data analysis and interpretation of findings revealed that smart contracts contribute to the process automation and digitalisation. The digitalised and self-executed process improves procurement efficiency. Meanwhile, smart contracts also elevate the level of trust and reliability between procurement participants. On the other hand, improved procurement efficiency and reliability cooperate and support each other to reach an advanced position. In addition to these relationships, efficiency and reliability have particular inputs to gain competitive advantages. To summarise, smart contract implementation supports enterprises in gaining competitive advantages through improved procurement efficiency and reliability.

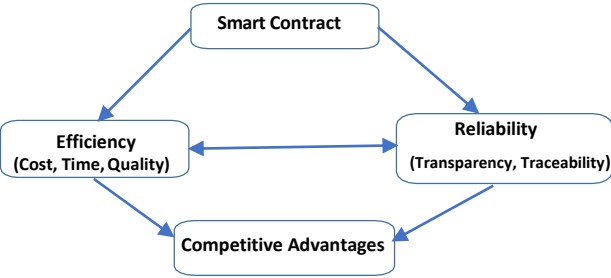

**Figure 3.** Relationships of concepts in an integrated environment.

## 5. Conclusions and Recommendations

This research sought to investigate the relationship between project procurement, competitive advantages, and the new phenomenon of smart contracts. Although previous researchers have studied the smart contract adaptation in financial services and supply chain management, they did not focus on how smart contracts can facilitate project management through project procurement. The research gap appeared in demonstrating the connection of smart contracts, project procurement, and competitive advantage concepts.

This study showed that project procurement is associated with supply chain management. Misconduct, ill-managed processes, low operation efficiency, and trust and security concerns were the major challenges discovered in the literature and collected data. The suggested solution to solve these problems was a smart contract implementation. Thus, the researcher initially researched the smart contract literature to understand this emerging technology. The academic and non-academic sources confirmed that smart contracts improve transparency, traceability, and data security. Furthermore, they implied the trust-less business environment built by smart contracts and Blockchain. The interview data analysis pointed out that smart contracts can deal with procurement challenges through increasing the procurement process efficiency and reliability among procurement participants. The automation and digitalisation of the process reduce the number of intermediaries, resource demand, and operating time. The decrease in the participant number, required resources, and task time creates result in cost-savings. Interviewees noted the benefits of smart contracts for elevating the data quality, information transparency, and traceability. These features of smart contracts address procurement problems, which are misconduct, ill-managed process, and trust and security concerns. The information quality correlates closely with the process and product quality.

Betterments in a procurement process through smart contracts equip enterprises to gain competitive advantages. Decreased operation costs can be used for cost leadership strategy, whilst the advancements in quality and database can take a role in the differentiation strategy. After reliability challenges are eliminated, buyer and supplier relationships can be much closer and more productive. Moreover, suppliers divest themselves of the bargaining power as a result of the information transparency. In addition, the early technology adaptation eases to implement future technologies. Smart contracts reinforce sustainable and ethical procurement. This can change customers' perceptions and attract their attention.

### 5.1. Implications for Research and Practices

This research aimed to reveal unaddressed knowledge and relationships which can inspire and be used in future research. This research added a different dimension into the Blockchain and smart contract literature. The significant contribution of this study to the literature is illuminating the correlation of project procurement management success, smart contracts, and competitive advantages. Not only advantages but also shortcomings of blockchain and smart contracts were mentioned in the presentation of findings. These experienced-based findings can be taken into consideration by other researchers to conduct research on these areas. Processes digitalisation, job loss, and internal resistance are momentous outcomes of smart contract implementation. This study can lay a foundation for future research which aims to investigate the interrelation of smart contracts and competitive advantages.

Individuals and companies make breakthroughs in internet technologies in every moment. Changing technology alters our lifestyles and demands. Business environments also evolve alongside technology. Even though tools and methods change, the desire of companies towards minimising costs and maximising profit never changes. Hence, enterprises need solutions to reduce their operational expenses. Project-based companies seek comprehensive technologies and solutions to resolve their business inefficiencies. This study can guide project-based companies to understand the possible outcomes of smart contract implementation. Companies that are dealing with a complex and an under-

performing supply chain or procurement process can have a broad idea as to how smart contracts and Blockchain work in supply chain and procurement. Undoubtedly, this study is important and value-added by reason of its contribution to the literature and business environment.

### 5.2. Research Limitations

During the entire research process, we faced several minor and major challenges. These internal and external challenges imposed limitations on this study. They are as follows:

- Blockchain and smart contract technologies are not well-researched. Reliable studies published on these topics are rare. The literature contains contradicting information. Therefore, we had a limited academic source to acquire sufficient information.
- Phenomenology seeks to understand the investigating phenomena through participants' experiences. The number of experts' works in the blockchain and smart contracts area are limited. Although some people work in this industry, they have not experienced a real-life project. Consequently, we had only a few experts to invite for the interviews and we were only able to attain limited experience-based information.
- A small number of smart contract implementations and constraints mentioned in point 3 restricted us to performing data triangulation by observations.

### 5.3. Recommendations for Future Research

Machine learning and artificial intelligence (AI) are phenomena in emerging technologies. Since the number of studies on these topics is increasing, futurists are expecting a world managed by machines. Machine learning and AI are data-driven technologies. Smart contracts and Blockchain can produce huge datasets. The amount of data can soar by the support of IoT devices. We believe that future research which combines smart contract, machine learning, and AI will make profound changes in the literature and industry practices.For example, the novel COVID-19 pandemic has taught us that we need to learn how to design automatic procurement processes. Future research should elaborate on this aspect for protecting companies to stop their operations. We also recommend to the Blockchain and smart contract community to investigate the challenges and opportunities of application of Blockchain-enabled smart contract is small-sized companies and businesses in the future. Lastly, machine learning and process automation are expected to cause job losses and internal resistance. This is a cultural change, and the cultural impacts of these technologies should be researched.

**Author Contributions:** Conceptualization, E.Ö. and O.H. methodology, E.Ö. and O.H.; validation, N.A.; formal analysis, E.Ö., N.A. and O.H.; investigation, N.A.; resources, N.A. and E.Ö.; data curation, E.Ö., N.A. and O.H.; writing—original draft preparation, E.Ö.; writing—review and editing, N.A. and O.H.; visualization, O.H.; supervision, O.H.; project administration, N.A. and O.H. All authors have read and agreed to the published version of the manuscript.

**Funding:** This research received no external funding.

**Institutional Review Board Statement:** Not applicable.

**Informed Consent Statement:** Not applicable.

**Data Availability Statement:** Not applicable.

**Conflicts of Interest:** The authors declare no conflict of interest.

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
