# Peer review of "Leveraging Smart Contract in Project Procurement through DLT to Gain Sustainable Competitive Advantages"

_sustainability, doi:10.3390/su132313380_

Round 1
Reviewer 1 Report
Given that smart contracts are made up of computer code, they are highly configurable and can be designed in many different ways, offering many types of services and solutions. Being decentralized and self-executing programs, smart contracts are able to provide greater transparency and lower operating costs than their traditional counterparts. Depending on the approach and implementation, they can also increase efficiency and reduce bureaucracy expenses.
Smart contracts turn out to be especially useful in situations that require the transfer of assets or the exchange of funds between two or more parties.
As of today, smart contracts already have many real examples of their use. Currently, they are most often used in the following areas: asset tokenization, voting systems, wallets for storing cryptocurrencies, decentralized exchanges and in the production of games and mobile applications. Smart contracts can also be simultaneously implemented with other solutions provided by blockchain technology in such areas as healthcare, charity, supply chain, broadly understood management or in the field of decentralized finance.
Smart contracts are certainly an interesting and breakthrough technology. However, what are their advantages - features such as dispersed nature, deterministic, transparent and in a sense non-modifiable - can very quickly turn into their disadvantages.
Considering the fact that smart contracts are composed of computer code written by people, this carries many risks. Any human-written code is potentially vulnerable to vulnerabilities and bugs. As a rule, smart contracts are created by experienced programmers, but as this sphere is gaining more and more popularity, it is worth being aware that poorly implemented code can lead to irreversible material losses.When criticizing smart contracts, it is worth emphasizing that smart contracts do not actually turn out to be such a great solution to many real problems. In fact, some organizations find it better and easier to use conventional centralized server-based alternatives than to base their solutions and systems on blockchain technology. Compared to smart contracts, today centralized servers turn out to be easier and cheaper to maintain. They also show higher efficiency. Another limitation of smart contracts is related to their uncertain legal status. Not only because this field is currently in the so-called gray zone in most countries, but mainly because smart contracts do not correspond to the legal framework presently in the world.Taking into account the widely described methodology and the great contribution of the authors to the process of analyzing small scientific literature and the small experience of the audited people, I believe that the reviewed article has a significant scientific contribution and its publication in this journal can be considered.I am asking the authors to present a specific example of the application of the proposed solution for an infrastructure contract in a given country.
Author Response
Our thanks for your comments. We have carefully addressed them all as follows:
- I am asking the authors to present a specific example of the application of the proposed solution for an infrastructure contract in a given country.
Response: Thanks for your comment. This has been addressed in page 8 by introducing a smart contract payment security system has been developed and implemented in a construction project in Turkey.
- Is the content succinctly described and contextualized with respect to previous and present theoretical background and empirical research (if applicable) on the topic? Reviewer 1 said: “can be improved”
Response: Thanks for your comment. We have discussed the details of this paper in the revised draft. Please see page 5, 6, 7 and 9.
- Are the arguments and discussion of findings coherent, balanced and compelling? For empirical research, are the results clearly presented? Reviewer 1 said: “can be improved”
Response: Thanks for your comment. We have discussed the details of this paper in the revised draft. Please see page 12 to 21.
- Is the article adequately referenced? Reviewer 1 said: “can be improved”
Response: we added the reference to smart contracts. Please see the reference section.
Thanks a lot.
Reviewer 2 Report
Smart contracts are still new topics, so there is lack of research information. So, we can identify new risk and disadvantages.
Anyway, I personally think, smart contracts will be used in the future more freqently and they have high potential.
I have questions for authors, which might be aswered in the imroved version:
- Critics say that coding can be so good as programmer is. So, how we can decrease the effects, if the code of the smart contract is wrong?
- How smart contracts can be ued in small sized companies?
- What happends if we need to change/cancel omething in procerument? It is possible to use this step in SC?
Author Response
Our thanks for your comments. We have carefully addressed them all as follows:
- Are the arguments and discussion of findings coherent, balanced and compelling? Reviewer 2 said: “can be improved”
Response: Thanks for your comment. We have discussed the details of this paper in the revised draft. Please see page 12 to 21.
- Is the article adequately referenced? Reviewer 2 said: “can be improved”
Response: we added the reference to smart contracts. Please see the reference section.
- Critics say that coding can be as good as programmer is. So, how we can decrease the effects, if the code of the smart contract is wrong?
Response: Thanks for your comment. The smart contract system is developed based on a rigorous audit system using dynamic and static auditing tools. We have addressed this concept in page 8.
- How smart contracts can be used in small sized companies?
Response: Thanks for your comment. Unfortunately, that’s not the scope of this research but application of smart contract in small sized companies and business would be a very interesting topic and we proposed it in the future direction section. Please see page 24 for more details.
- What happens if we need to change/cancel something in procurement? It is possible to use this step in SC?
Response: Thanks for your comment. This issue can be addressed by a dynamic service-level agreement (SLA) enabled on a two-layer blockchain architecture. Please see page 19 for more info and details.